# Involvement of *Helicobacter pylori* in Preoperative Gastric Findings on a Bariatric Population

**DOI:** 10.3390/ijerph19159088

**Published:** 2022-07-26

**Authors:** Soledad García-Gómez-Heras, María Jesús Fernández-Aceñero, Gilberto González, María de Lourdes Bolaños-Muñoz, Raquel Franco-Rodríguez, Julio Paredes-González, Jaime Ruiz-Tovar

**Affiliations:** 1Department of Basic Health Science, Health Science Faculty, Rey Juan Carlos University, 28922 Madrid, Spain; raquel.franco@urjc.es (R.F.-R.); julio.paredes@urjc.es (J.P.-G.); 2Department of Pathology, Hospital Clínico Universitario San Carlos, 28040 Madrid, Spain; mgg10167@gmail.com; 3Department of Surgery and Bariatrics, Centro Médico Puerta de Hierro, Guadalajara 45040, Mexico; gilpchmd@yahoo.com.mx; 4Neuroscience Institute CUCBA, Guadalajara University Guadalajara, Guadalajara 44340, Mexico; psic_lourdesb@yahoo.com.mx; 5Department of Health Sciences, Alfonso X El Sabio University, 28691 Madrid, Spain; jruiztovar@gmail.com

**Keywords:** *Helicobacter pylori*, bariatric surgery, gastritis, gastric atrophia, gastric metaplasia

## Abstract

The prevalence of *Helicobacter pylori* (Hp) in bariatric patients is common and related to gastric pathology. With preoperative upper gastrointestinal endoscopy (UGE), these pathologies and the presence of Hp are diagnosed. The histopathological study of the UGE biopsies is classified based on the Sydney System, a scoring system that stages chronic gastritis (CG) and precancerous gastric lesions. The objective is to assess the histological findings of gastric biopsies during routine UGE and to determine the involvement of Hp in gastric disorders in patients undergoing bariatric surgery. A multicenter retrospective review of prospectively collected databases was performed. The presence of CG, gastric atrophy (GA), and gastric intestinal metaplasia (GIM) in the study of the biopsies was assessed and correlated with Hp infection. The incidence of Hp among our bariatric population was 36.1%, and it increases with age. The percentage of patients with severe Hp infection is higher in patients with GA or GIM. The Hp eradication rate is also reduced when GA and GIM are present. A histological examination of all the biopsies did not show features of malignancy in any of the cases. Hp is not the only factor involved in the development of gastric pathology in bariatric patients.

## 1. Introduction

Bariatric surgery is considered the gold standard treatment for morbid obesity, as it achieves a significant and maintained change in body mass index (BMI) and improvement of comorbidities, such as type II diabetes mellitus, hypertension, non-alcoholic fatty liver disease (NAFLD), sleep apnea syndrome, etc., [1]. Sleeve gastrectomy (SG) and Roux-en-Y gastric bypass (RYGB) are the two most frequently performed procedures. Both lead to anatomic modifications of the stomach and the small bowel [2,3,4,5,6].

Different guidelines recommend an endoscopic examination of the digestive tract before surgery so that the surgeon can be aware of any pathology, either symptomatic or not, that may need treatment before the surgical procedure or that may need follow-up in the postoperative course. However, there is still a certain controversy regarding the routine indication of preoperative upper gastrointestinal endoscopy (UGE) [3,7,8].

One indication for conducting a UGE is to determine gastric *Helicobacter pylori* (Hp) infection. This infection is very common in morbidly obese patients, with a reported higher prevalence than in the lean population. In the last few years, epidemiological studies revealed that the prevalence of Hp in bariatric patients ranged from 23% to 70% [9]. In the regions where the authors work (Spain and Mexico), the prevalence of Hp infection in the morbidly obese population also varies widely between studies (17.5–69.5%) [1,10,11].

The eradication of Hp before the bariatric approach is mandatory, as the infection has been associated with postoperative complications [9]. In addition, Hp has been related to gastric pathologies (Figure 1), such as peptic ulcers, chronic gastritis, gastric atrophy, intestinal metaplasia, and gastric adenocarcinoma. Thus, despite recent challenges that have shown the possibility of access and intervention on the blind-ended gastric remnant [12], it is not recommended to perform bariatric techniques with gastric exclusion in patients without prior eradication of Hp, as a part of the stomach would remain with difficult access for postoperative endoscopic control [12].

In 1994, the International Agency for Research on Cancer considered Hp as a carcinogenic agent for type I gastric cancer (GC). It is important to emphasize that more than 90% of patients with GC suffer from an active or past infection by Hp [13]. According to the model of the pathogenesis of stomach carcinogenesis proposed by Correa [14], the sequential changes in the gastric mucosa are: first chronic gastritis (CG), evolving to gastric atrophy (GA), and then gastric intestinal metaplasia (GIM), which could eventually lead to GC. These clinical situations are often associated with an Hp infection [13,14,15].

However, Hp infection is not the only factor involved in the development of gastric pathology. Other factors, such as alcohol intake, tobacco habits, or inadequate diet, among others, have also been involved in carcinogenesis [13].

The present study aimed to assess the histological findings of gastric biopsies during routine UGE and to determine the involvement of Hp in histological gastric disorders in patients undergoing bariatric surgery.

## 2. Patients and Methods

### 2.1. Selection of Patients

A retrospective review of prospectively collected databases of all UGE performed before primary bariatric surgery between 2012 and 2019 at two university urban hospitals in Spain and Mexico was performed. Information was recorded from the revision of clinical histories.

UGE was routinely performed on all the patients as part of the preoperative evaluation protocol. Endoscopic biopsy sites were standardized; two biopsies were taken from the antrum (along the lesser and greater curvatures), one biopsy from the incisura angularis, and two biopsies were taken from the gastric body (along the lesser and greater curvatures).

All gastric pathologies were diagnosed by UGE with biopsy and histopathological study, whose classification was based on the Sydney System (SS) [16], a scoring system protocol that stages the GA and the GIM as precancerous lesions, and the CG. The SS includes endoscopic and histological aspects.

### 2.2. Description of the Histological Examination and Recorded Variables

For histological studies, the tissue samples of 5 mm^3^ were fixed in 10% formaldehyde at room temperature, embedded in paraffin, and cut into five-micron sections in a Micron HM360 microtome. Sections were stained with hematoxylin-eosin. All were diagnosed with a Zeiss Axiophot 2 microscope following the usual standardized report. Biopsies were evaluated according to the SS key parameters: etiology, topography, morphology, and grade (Figure 2).

The presence of Hp was assessed by the CLO test (MRM Comercial, Valladolid, Spain). The CLO test is a rapid urease test (RUT) that accurately detects the presence of the urease enzyme and the presence of the Hp urease enzyme in gastric mucosal biopsies. The CLO test consists of a well of urease indicator gel sealed in a plastic plate which, when in contact with a tissue biopsy, is at least 2 cm away from the biopsy site in the pylorus. If Hp urease is present in the sample, the gel will change from yellow to deep magenta. The gel is composed of urea, phenol red (a pH indicator), buffer solutions, and a bacteriostatic agent to prevent the growth of urease-positive organisms.

### 2.3. Selection Criteria of the Surgical Technique

The criteria for performing bariatric surgery were a body mass index (BMI) ≥ 40 kg/m^2^ or a BMI ≥ 35 kg/m^2^ with inadequately controlled obesity-related comorbidities (e.g., T2D, hypertension, dyslipidemia, NAFLD, or sleep apnea/hypopnea syndrome). Patients with a BMI of over 50 kg/m^2^, high surgical risk due to comorbidities (American Society of Anesthesiology Classification Stage IV (ASA IV), comorbidities requiring chronic medication (such as corticoids, immune suppressors, etc.), intestinal inflammatory disease, and a probability of technical difficulty before surgery (multiple previous abdominal surgeries or known intra-abdominal anatomic modifications) were assigned to the SG procedure. In patients with clinical symptoms of gastroesophageal reflux disease (GERD) or endoscopic findings suggestive of esophagitis, RYGB was considered the bariatric technique of choice. Those patients presenting endoscopic gastric or duodenal findings, that might require an eventual endoscopic follow-up (ulcers, partially resected polyps, or adenomatous polyps) were selected for SG. Similarly, the histologic findings of GA or GIM, considered premalignant lesions, were indications of SG. 

The cases with a positive diagnosis of Hp were treated with diverse therapeutic schemes to achieve preoperative eradication; bismuth subcitrate 140 mg, metronidazole 125 mg, and tetracycline hydrochloride 125 mg schemes (Pylera, Allergan) were used as first-line treatments. If eradication was not achieved, second and third-line treatments were prescribed. If eradication could not be finally achieved, the patient was selected for SG or, if a GERD diagnosis coexisted, for RYGB with resection of the gastric remnant. Hp eradication after treatment was assessed by a breath test (rapid urease test). Routine endoscopic follow-up of the gastric lesions was not performed as those premalignant lesions or findings amenable to endoscopic follow-up were selected for SG or resection of the gastric remnant. Elective endoscopic follow-up was performed during the postoperative course in symptomatic patients or those without eradication of Hp.

In the rest of the cases, without any of the previously mentioned conditions, the surgical technique was decided in the weekly meetings of the multidisciplinary team, customizing the decision based on age, comorbidities, and the expected results with each approach. The recommendation of the multidisciplinary team was communicated to the patients and their opinion was also considered for the final decision.

### 2.4. Variables

The Hp infection and its grade were determined. The presence of chronic gastritis (CG), gastric atrophy (GA), and gastric intestinal metaplasia (GIM) in the histopathological study of the biopsies was assessed. These entities were classified based on the Sydney System and correlated with Hp infection.

### 2.5. Statistical Analysis

Data analysis was performed using SPSS Version 22.0 (SPSS Inc., Chicago, IL, USA). Results are expressed as means and standard deviation (SD) or numbers and percentages. Qualitative variables were compared with the Chi-squared test. A *p*-value < 0.05 was considered statistically significant.

## 3. Results

A total of 1040 patients undergoing bariatric surgery were included. All of them underwent a preoperative UGE with gastric biopsies. 

As shown in Table 1, the mean age of the studied population was 45.5 + 10.8 years, most of the females (67%) and over 50 years old (67%). The mean preoperative BMI was 46.8 + 8.9 kg/m^2^, with 62.5% of the patients within the range of 40–49 kg/m^2^, and 16.9% over 50 kg/m^2^. RYGB was the most frequently performed technique (77.9%) (Table 2). The type of surgical procedure does not affect the Hp infection or eradication effectivity. Postoperative complications appeared in 18 patients (1.7%): 3 staple line leaks, 3 incisional surgical site infections, 1 hemoperitoneum after SG, 4 gastro-jejunal anastomotic leaks, 3 hemoperitoneum, 3 incisional surgical site infections, and 1 entero-enteral anastomotic leak after RYGB.

The analysis of Hp showed that 375 patients (36.1%) presented with preoperative Hp infection. After antibiotic therapy (before bariatric surgery), Hp was eradicated in 94.7% of the cases. Those cases without preoperative eradication (20 patients) were selected preferentially for Sleeve gastrectomy (SG) or, in the case of coexistence of gastroesophageal reflux disease (GERD), a Roux-en-Y gastric bypass (RYGB) with the removal of the gastric remnant. The lack of Hp eradication was not associated with increased postoperative complication rates. The intensity of Hp infection is summarized in Table 3.

The Hp infection rate was lower in patients with greater BMI: Hp infection rate was 44.4% in patients with BMI between 35–40 Kg/m^2^, whereas the infection rate decreased to 34.8% in patients with BMI of 40–50 Kg/m^2^ and 30.4 Kg/m^2^ when the BMI is over 50 Kg/m^2^ (*p* = 0.006). The Hp infection rate was significantly greater in patients over 50 years (*p* = 0.001), without the influence of gender. Postoperative complications did not show any significant correlation with Hp infection or eradication.

### Prevalence of CG, GA, and GIM and Their Association with Hp

The prevalence of CG, GA, and GIM among our bariatric population was 47%, 3.6%, and 7.7%, respectively. All the patients with the diagnosis of Hp infection presented with gastritis (100%). However, Hp infection was only present in 76.7% of the cases of CG. Similarly, 4.5% of the patients with Hp infection presented GA, but Hp accounts for only 44.7% of the cases of GA. Finally, 8.5% of the patients with Hp infection presented GIM, but Hp was involved in only 40% of the cases. In those cases with Hp infection, we observed that the greater the grade of Hp infection, the higher the rate of gastric atrophy and metaplasia (Table 4). Despite the greatest Hp infection rate among patients over 50 years old, the global CG, GA, and GIM showed no significant differences depending on age (Table 5). Similarly, Hp infection rates significantly decreased in patients with higher BMI, but significant differences in the infection severity grade according to BMI could not be determined. Eradication rates remained similar in all the groups, without the relevant influence of BMI (Table 6). A histological examination of all the biopsies did not show features of malignancy in any of the cases.

As previously mentioned, the global eradication rate of Hp in the analyzed sample was 94.7% (355 out of 375 patients with Hp infection). However, in those cases with Hp infection and GA (17 cases), the eradication rate was only 47.1% (8 eradicated cases out of the 17 with Hp infection). Similarly, in those patients with Hp infection and GIM, the eradication rate decreased to 37.8% (12 eradicated cases out of the 32 with Hp infection). 

Postoperative complications did not show any significant correlation with the presence of GA or GIM.

## 4. Discussion

The routine performance of UGE may lead to minimizing complications in the postoperative period, especially related to ulcer perforations or the development of malignancies in ulcers or polyps [17,18,19].

Thus, different international guidelines recommend a preoperative UGE, mostly in patients with GERD or symptomatic gastric disorders. The Spanish guidelines for bariatric surgery [20] recommend performing a routine UGE or contrast-enhanced esophagus-gastro-duodenal series only in those patients who will undergo a procedure including a gastric remnant and duodenal exclusion. The European Association for Endoscopic Surgery (EAES) [21] recommends one of these complementary tests for all patients in the preoperative assessment.

The main drawback of these recommendations is that clinical symptoms are present in less than 20% of the cases, and contrast-enhanced series have lower accuracy than UGE for the diagnosis of gastric pathology and do not allow for the acquisition of biopsies so histological findings remain unknown [8]. 

Several studies have demonstrated that Hp is the most frequent factor leading to gastric pathology (erosions, ulcers, perforations, neoplasms, etc.) In addition, it has been associated with postoperative complications. The eradication of H. pylori has been associated with a reduction of all these entities. Diverse Hp-related gastric lesions are at risk of developing gastric cancer and are consequently considered premalignant lesions (GA, GIM, etc.) [13,14,15]. Moreover, gastric disorders are often at risk of bleeding, and having endoscopic access to control them is advisable. Consequently, given the risk of complications related to gastric disorders, it seems mandatory to diagnose them preoperatively, treat them if possible, and guarantee endoscopic access to them, to control their eventual evolution to neoplasms or the appearance of complications (perforation or bleeding) [22]. Hp infection can be diagnosed with techniques other than a biopsy (urease test). However, the endoscopic and even more relevant, histologic implications of the Hp infection remain unknown when UGE is not performed.

A relevant finding of the present study is that Hp is not the only factor involved in the development of gastric pathology. In our series, Hp accounts for only 76.7% of gastritis, 44.7% of GA, and 40% of GIM. This reveals that other factors are present. Significant associations have been established with alcohol intake, tobacco habit, biliary reflux, and inadequate diets, with low fruit and vegetable intake [23]. In a morbidly obese population, the latter is probably the most associated parameter with GA and GIM development, once discarded Hp infection. Consequently, gastric pathology cannot be discarded with just Hp eradication. Thus, we would recommend performing a routine UGE with mucosal biopsy to discard not only Hp infection (which can be established with other methods), but the presence of GA or GIM, which are eventually present in the absence of Hp infection. Contrast-enhanced gastric series, supported by several scientific associations [20,21], may misdiagnose microscopic findings in the absence of macroscopic lesions evident in radiological examinations.

Furthermore, it has been observed that the percentage of cases with severe Hp infection tends to be higher in patients with GA or GIM than in patients without these lesions. The relationship between H. pylori density and GA and GIM has been previously reported [24,25]. Probably, in those cases of GA and GIM with Hp involvement, both the bacterial load and the genotype of *Helicobacter* can influence the changes found in the histopathologic analysis and the patients’ outcome [26]. Moreover, Hp density has not only been associated with the presence of GA and GIM but also with their activity [24]. GA and GIM activity were not determined in the present study, but it could be interesting to analyze them in future projects. In contrast to these results, there is also evidence that Hp infection may decrease as the degree and severity of gastric inflammation increases, being less frequent in more advanced premalignant lesions [27,28]. With these data, several authors suggest that HP infection is a necessary but not sufficient cause for the development of gastric adenocarcinoma since Hp can spontaneously disappear during gastric adenocarcinoma development in infected patients. Other authors consider that this gradual decrease in the concentration of the bacterium is because the most severe lesions of the carcinogenic sequence would be an unfavorable and inhospitable environment for the growth of Hp, even disappearing in gastric adenocarcinoma. Therefore, Hp, in particular, is considered to have an important role as an initiator and promoter of the carcinogenic cascade, but, because the density of bacterial colonization decreases in subsequent lesions, the subsequent progression of premalignant lesions is less dependent on Hp and more related to other environmental and genetic factors of the host [29,30,31,32].

On the other hand, it is remarkable that the Hp eradication rate is significantly lower when GA and GIM are present. It has already been determined that eradication resistance is closely related to Hp density. This is especially relevant for the development of Hp resistance to Claritromicine [33]. It seems to be logical that when GA or GIM are present, there is a greater density of Hp and, consequently, the development of antibiotic resistance is more frequent. However, it should also be elucidated if this resistance might be associated with the inflammatory response of the organism to GA and GIM lesions, which might prevent an adequate effect of the antibiotic treatment.

The global incidence of Hp infection among our bariatric population was 36.1%, and this rate increases with age, as already reported by other authors. However, this incidence is relatively low when compared with other series reporting a prevalence reaching 70% of the population [9]. Notwithstanding, the prevalence rate in different studies is very variable, ranging from 17–70%, even in the same populations [1,10,11]. It can be assumed that obesity is a risk factor for Hp infection, probably secondary to the certain immune deficiency related to the proinflammatory status associated with obesity. Surprisingly, we observed that the Hp infection rate was lower in the group of patients with greater BMI. The reason for this finding remains unknown to us, and further studies should be conducted to confirm these findings. 

## 5. Limitations of the Study

In this study, the presence of Hp was assessed by a rapid urease test (RUT), detecting the presence of the urease enzyme and the presence of Hp urease enzyme in gastric mucosal biopsies. Most current guidelines recommend that the detection of Hp must be determined based on at least two positive tests [34]. As this is a retrospective review of patients managed between 2012 and 2019, the detection of Hp was performed according to the routine institutional protocol at that time. Thus, the Hp infection determination might not be accurate enough, when compared with the actual guidelines.

As this is a retrospective study, confounding factors associated with CG, GA, and GIM, such as alcohol intake, diet, tobacco habit, or gastroesophageal reflux symptoms, could be present but were not determined for this study.

Further prospective studies must evaluate the role of Hp, in combination with other causal factors, in the development of CG, GA, and GIM.

## 6. Conclusions

The prevalence of Hp infection in a bariatric population reaches 36.1%. The prevalence of CG, GA, and GIM among our bariatric population was 47%, 3.6%, and 7.7%, respectively. Despite the fact that Hp is involved in the development of CG, GA, and GIM, it does not account for all the cases of these gastric pathologies. In those cases, with Hp infection, the greater the grade of Hp infection, the higher the rate of GA and GIM. In addition, the eradication rate of Hp significantly decreases when GA or GIM are present.

## Figures and Tables

**Figure 1 ijerph-19-09088-f001:**
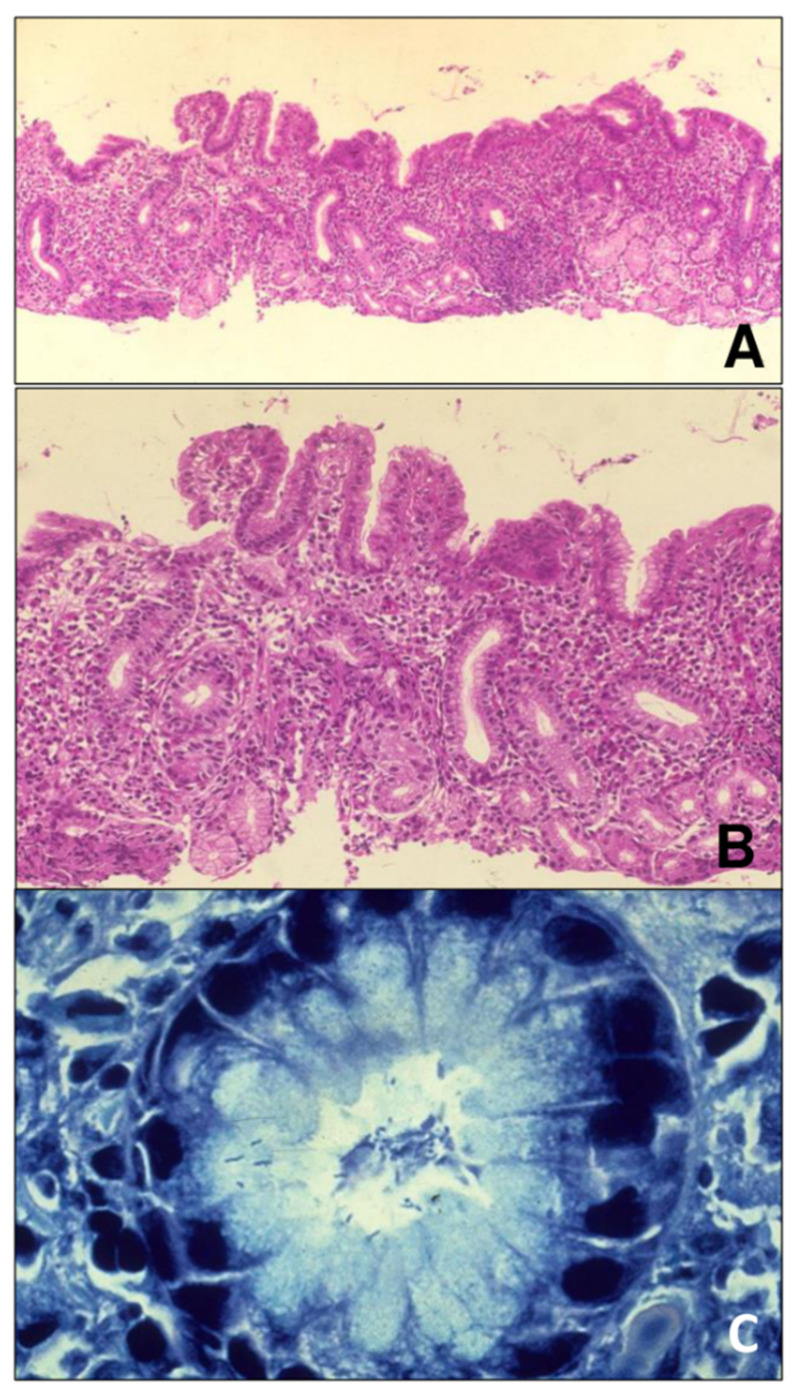
(**A**). Antral mucosa with inflammation, atrophy, and lymphoid follicles. Hematoxylin-eosin x40. (**B**). Same image as in (**A**), but at higher magnification. Chronic gastritis with inflammation infiltrate around the crypts and plasm cells. Hematoxylin-eosin, x100. (**C**). Glands with *Helicobacter pylori* Giemsa, x400.

**Figure 2 ijerph-19-09088-f002:**
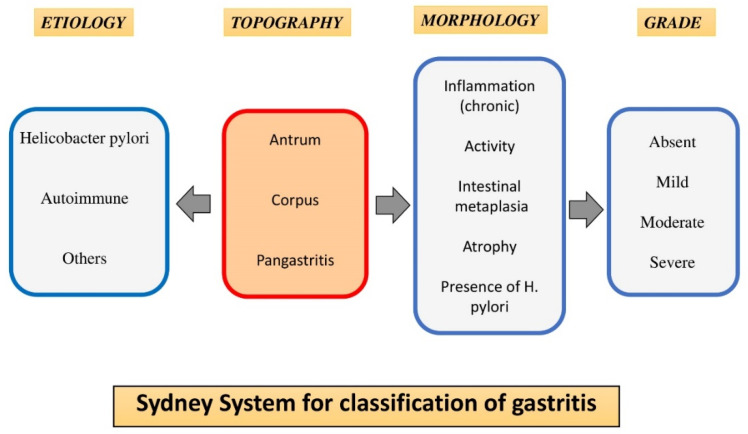
Scheme of Sydney System protocol.

**Table 1 ijerph-19-09088-t001:** Demographic features of the patients.

	N (1040)
**Age (years old)**	
**Mean ± SD**	45.4 ± 10.8
**Age grouped**	
**≤50**	697
**>50**	343
**Gender**	
**Male**	344
**Female**	696
**BMI Kg/m^2^**	
**Mean ± SD**	46.8 ± 8.9
**35–39.9 kg/m^2^**	214
**40–49.9 kg/m^2^**	650
**≥50 kg/m^2^**	176
**Total**	1040

BMI: Body mass index; SD: Standard deviation.

**Table 2 ijerph-19-09088-t002:** Type of surgical procedures according to gender and age.

	N (%)	≤50 Years Old N (%)	>50 Years Old N (%)	Male N (%)	Female N (%)
**Roux-en-Y gastric bypass (RYGB)**	810 (77.9%)	231 (28.5%)	579 (71.5%)	255 (31.5%)	555 (68.5%)
**Sleeve Gastrectomy (SG)**	230 (22.1%)	66 (28.7%)	164 (71.3%)	105 (45.7%)	125 (54.3%)

**Table 3 ijerph-19-09088-t003:** Rate of *Helicobacter pylori* infection in the series.

	N (1040)	%
**Hp positive**	375	36.1
**Mild Hp**	105	28
**Moderate Hp**	158	42.1
**Severe Hp**	112	30.9
**Hp eradicated**	355	94.7

Hp: *Helicobacter pylori*.

**Table 4 ijerph-19-09088-t004:** *Helicobacter pylori* in the population according to gender, age, and histopathologic findings in the pre-surgery biopsy.

	≤50 Years N (%) 697 (67%)	>50 Years N (%) 343 (33%)	*p* *	Female N (%) 696 (67%)	Male N (%) 344 (33%)	*p* *	Gastritis N (%) 489 (47%)	Atrophy N (%) 38 (3.6%)	Metaplasia N (%) 80 (7.7%)
***H. pylori* +** **375 (36.1%)**	202 (29%)	173 (50.4%)	0.001	263 (37.8%)	112 (32.6%)	NS	375 (76.7%)	17 (44.7%)	32 (40%)
**Mild** **105 (28%)**	59 (29.2%)	46 (26.6%)	NS	75 (28.5%)	30 (26.8%)	NS	105 (28%)	4 (23.5%)	0
**Moderate** **158 (42.1%)**	80 (39.6%)	78 (45.1%)		115 (43.7%)	43 (38.4%)		158 (42.1%)	6 (35.3%)	14 (43.8%)
**Severe** **112 (30.9%)**	63 (31.2%)	49 (28.3%)		73 (27.8%)	39 (34.8%)		112 (30.9%)	7 (41.2%)	18 (56.2%)
***H. pylori* eradicated** **355 (94.7%)**	193 (95.5%)	162 (93.6%)	NS	252 (95.8%)	103 (92%)	NS	355 (94.7%)	8 (47.1%)	12 (37.8%)

*p* * significant (S) or non-significant (NS).

**Table 5 ijerph-19-09088-t005:** Prevalence of gastritis, atrophy, and metaplasia in the population stratified by age.

	≤50 Years N (%) 697 (67%)	>50 Years N (%) 343 (33%)	*p* *
**Gastritis** **N (%) 489(47%)**	285 (40.9%)	132 (38.5%)	NS
**Atrophy** **N (%) 38 (3.6%)**	26 (3.7%)	12 (3.5%)	NS
**Metaplasia** **N (%) 80 (7.7%)**	48 (6.9%)	34 (9.9%)	NS

*p* * significant (S) or non-significant (NS).

**Table 6 ijerph-19-09088-t006:** *Helicobacter pylori* in the population according to BMI (kg/m^2^).

	BMI 35–40 N (%) 214 (20.6%)	BMI 41–50 N (%) 650 (62.5%)	BMI >50 N (%) 176 (16.9%)	*p* *
***H. pylori* +** **375 (36.1%)**	95 (44.4%)	226 (34.8%)	54 (30.7%)	0.006
**Mild** **105 (28%)**	20 (21.1%)	74 (32.8%)	11 (20.4%)	NS
**Moderate** **158 (42.1%)**	55 (57.8%)	76 (33.6%)	27 (50%)	NS
**Severe** **112 (20.9%)**	20 (21.1%)	76 (33.6%)	16 (29.6%)	NS
***H. pylori* eradicated** **355 (94.7%)**	90 (94.7%)	214 (94.7%)	51 (94.4%)	NS

*p* * significant (S) or non-significant (NS). BMI: body mass index.

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
