# Peer review of "Involvement of Helicobacter pylori in Preoperative Gastric Findings on a Bariatric Population"

_ijerph, 2022, doi:10.3390/ijerph19159088_

Round 1

Reviewer 1 Report

This submission is in poor English and extremely suffers from several grammatical, punctuation and typo errors though the manuscript text.

- Introduction: needs to be revised to better present the significance and main goal of the study.

- Methods: the presence of Hp was assessed only by CLO test, which is a rapid urease test (RUT). This is not adequate to detect H. pylori infection as recommended by most guidelines. The detection of Hp must be determined based on at least two positive tests recommended by Hp management guidelines.

- Results: this section must be divided up by relevant subsections to easily present the findings.

- Discussion: is short and not sufficiently addressed the main findings. More citations must be used to compare and interpret the results in this part.

- Limitations of the study: this part is partial and not accurate. It is better to directly mention the main limitations.

- Correct “rapid urease test (TRU)” to “rapid urease test (RUT)”, etc…

Author Response

Dear Reviewer 1, 

Dr. García-Gómez-Heras

(Corresponding Author)

Reviewer 2 Report

This article discusses the histological findings of gastric biopsies during routine UGE and to assess the involvement of Hp in the histological gastric disorders in patients undergoing bariatric surgery. However, some modifications should be revised before accepting for publication.

1.       The objectives in the manuscript were to evaluate the histological findings of gastric biopsies in routine UGE and to determine the involvement of Hp in histological gastric disease in patients undergoing bariatric surgery. But the conclusions drawn to have no deterministic results, whether this is reasonable.

2.       It is concluded in the manuscript that Hp is not the only factor in the development of gastric pathology in obese patients, so for other possible factors, do the authors have some opinions of their own, please add.

3.       Insufficient Discussion section in the manuscript suggests adding discussion.

4.       Only some data on Hp and gastric-related diseases in obese patients are statistically analyzed in the manuscript, but the possible reasons for these phenomena are not discussed and suggested to be added.

5.       It is suggested to revise Figure 1 in the manuscript to reduce the blank area, increase the clarity, and make the picture size consistent, and the table should be changed to a three-line table.

6.       It is recommended to pay attention to some details in the manuscript, such as the space for the legend in Figure 1 on line 53; the overall manuscript is to the right; the blank area after line 152; whether the Table font is bold, etc.

Author Response

Dear Reviewer 2, 

Dr. García-Gómez -Heras.

(Corresponding author)

Reviewer 3 Report

This study signifies the importance of increasing H. pylori infections and the manuscript is suitable for the publication in IJERPH. However, I have a concern with the Fig 1. The authors have provided the images of the same section for Panel  A and B but at a different magnification levels. The authors need to address the same, if they intended to use the same sections for panel A and B, they are suggested to provide an explanation for the same in the text. 

Author Response

Dear Reviewer 3,

Dr. García-Gómez-Heras

(Corresponding Author)

Round 2

Reviewer 1 Report

The authors have implemented the required corrections into the manuscript text.

Good luck.